# Clinical and biochemical features of atherogenic hyperlipidemias with different genetic basis: A comprehensive comparative study

Anastasia V. Blokhina[1]*, Alexandra I. Ershova[1], Anna V. Kiseleva[2], Evgeniia A. Sotnikova[2], Anastasia A. Zharikova[2,3], Marija Zaicenoka[2,4], Yuri V. Vyatkin[2,5], Vasily E. Ramensky[2,3,6], Vladimir A. Kutsenko[7], Olga A. Litinskaya[8], Maria S. Pokrovskaya[9], Svetlana A. Shalnova[10], Alexey N. Meshkov[2,11,12,13], Oxana M. Drapkina[14]

1 Laboratory of Clinomics, National Medical Research Center for Therapy and Preventive Medicine of the Ministry of Healthcare of the Russian Federation, Moscow, Russia, 2 Institute of Personalized Therapy and Prevention, National Medical Research Center for Therapy and Preventive Medicine of the Ministry of Healthcare of the Russian Federation, Moscow, Russia, 3 Faculty of Bioengineering and Bioinformatics, Lomonosov Moscow State University, Moscow, Russia, 4 Moscow Center for Advanced Studies, Moscow, Russia, 5 Department of Natural Sciences, Novosibirsk State University, Novosibirsk, Russia, 6 Institute for Artificial Intelligence, Lomonosov Moscow State University, Moscow, Russia, 7 Laboratory of Biostatistics, Department of Epidemiology of Chronic Non-Communicable Diseases, National Medical Research Center for Therapy and Preventive Medicine of the Ministry of Healthcare of the Russian Federation, Moscow, Russia, 8 Clinical Diagnostic Laboratory, National Medical Research Center for Therapy and Preventive Medicine of the Ministry of Healthcare of the Russian Federation, Moscow, Russia, 9 Biobank, National Medical Research Center for Therapy and Preventive Medicine of the Ministry of Healthcare of the Russian Federation, Moscow, Russia, 10 Department of Epidemiology of Chronic Non-Communicable Diseases, National Medical Research Center for Therapy and Preventive Medicine of the Ministry of Healthcare of the Russian Federation, Moscow, Russia, 11 National Medical Research Center for Cardiology of the Ministry of Healthcare of the Russian Federation, Moscow, Russia, 12 Hereditary Metabolic Diseases Laboratory, Research Centre for Medical Genetics, Moscow, Russia, 13 Department of General and Medical Genetics, Pirogov Russian National Research Medical University, Moscow, Russia, 14 Department of Fundamental and Applied Aspects of Obesity, National Medical Research Center for Therapy and Preventive Medicine of the Ministry of Healthcare of the Russian Federation, Moscow, Russia

* blokhina0310@gmail.com

**Data Availability Statement:** The data used in this study, including individual genotype information,

## Abstract

Patients with genetically-based hyperlipidemias exhibit a wide phenotypic variability. Investigation of clinical and biochemical features is important for identifying genetically-based hyperlipidemias, determining disease prognosis, and initiating timely treatment. We analyzed genetic data from 3374 samples and compared clinical data, lipid levels (low-density lipoprotein cholesterol (LDL-C), high-density lipoprotein cholesterol, triglycerides, and lipoprotein (a)), frequency, age at onset of coronary heart disease (CHD), and the severity of carotid and femoral atherosclerosis (plaque number, maximum stenosis, total stenosis, maximum plaque height, and plaque score) among patients with familial hypercholesterolemia (FH), familial dysbetalipoproteinemia (FD), polygenic hypercholesterolemia (HCL), severe HCL, and those without lipid disorders (n = 324). FH patients exhibited the highest LDL-C (median 8.03 mmol/L, p < 0.001). FD patients had elevated triglyceride levels (median 4.10 mmol/L), lower LDL-C (median 3.57 mmol/L), and high-density lipoprotein cholesterol (median 1.03 mmol/L) compared to FH, polygenic HCL, and severe HCL, p <

cannot be publicly disclosed according to the rules of the Ethics Committee of the National Medical Research Center for Therapy and Preventive Medicine. Deidentified data will be provided upon reasonable request by the corresponding author (blokhina0310@gmail.com) or by the researcher of the laboratory of clinomics, Dr. Limonova Alena (limonova-alena@yandex.ru) or by the Ethics Committee of the National Medical Research Center for Therapy and Preventive Medicine (phone number +7 (499)-553-68-10, SecretaryNEC@gnicpm.ru). Proposals will be reviewed and approved by the investigators, local regulatory authorities, and the Ethics Committee of the National Medical Research Center for Therapy and Preventive Medicine. Once the proposal is approved, data can be transferred through a secure online platform after signing a data access agreement and a confidentiality agreement.

**Funding:** The author(s) received no specific funding for this work.

**Competing interests:** The authors have declared that no competing interests exist.

0.05. FH and FD patients had similar early onset of CHD, with a median age of 44 and 40 years and comparable frequencies of 29.5% and 31.0%, respectively. They were more likely to develop CHD than subjects without lipid disorders ($p = 0.042$ and $p < 0.001$, respectively). Additionally, FH patients had higher a carotid plaque number, total carotid stenosis, and carotid plaque score. This study presents the first simultaneous comparison of clinical and biochemical features among FD, FH, polygenic, and severe HCL, along with the first comprehensive evaluation of carotid and femoral atherosclerosis ultrasound parameters in FD patients. The results highlight distinct phenotypic features unique to each hyperlipidemia analyzed and underscore FH and FD as the most atherogenic hyperlipidemias.

## Introduction

Hyperlipidemia, a major factor in the development of atherosclerosis, remains prevalent worldwide [1]. Elevated levels of lipoproteins may be attributed to lifestyle factors, comorbidities, a genetic basis, or a combination of these [1]. The early development of atherosclerosis is usually associated with inherited hyperlipidemias, which have a different genetic basis.

Familial hypercholesterolemia (FH) is one of the most common genetically-based hyperlipidemias, affecting approximately 1 in 173 to 300 individuals in the general population [2–4]. FH is characterized by early-onset atherosclerotic cardiovascular disease (CVD) and an increased risk of mortality due to the lifelong elevated low-density lipoprotein cholesterol (LDL-C) levels [1, 5]. FH is mainly caused by pathogenic or likely pathogenic variants in the low-density lipoprotein receptor (*LDLR*), the apolipoprotein B (*APOB*), and the proprotein convertase subtilisin/kexin type 9 (*PCSK9*) genes [6]. The severe phenotypic variability in patients with FH is determined by a combination of genetic diversity and a wide range of risk factors for atherosclerotic CVD. Subjects with various causal genes [7], a spectrum of genetic variants within the same gene [8], or even the same variants [9] demonstrate extremely variable LDL-C levels. In addition, FH may be compounded by hyperlipoproteinemia (a) [10, 11] and a high polygenic risk [12], which together influence both the variability of LDL-C levels and the risk of developing atherosclerotic CVD.

Familial dysbetalipoproteinemia (FD) is a highly atherogenic lipid disorder with a genetic basis. The prevalence of FD has been shown to be high, ranging from 0.2% to 2.7% [13–16], and may even be comparable to FH [13]. FD presents a complex multifactorial phenotype that could be difficult for timely diagnosis [17–19]. In over 90% of cases, the apolipoprotein E (*APOE*) ε2ε2 haplotype predisposes to the development of FD [20]. However, in contrast to FH, a genetic basis alone is generally not sufficient. FD has a delayed onset and requires additional factors involving overweight, obesity [21–23], insulin resistance [21, 22, 24], diabetes mellitus [21, 25], hypothyroidism, some medications, menopause [21] or pregnancy [26, 27], and so on. All of these factors contribute to the accumulation of cholesterol-enriched remnant lipoproteins, which are associated with an increased risk of premature atherosclerosis [28, 29].

Elevated LDL-C levels may also be associated with polygenic causes. The cumulative impact of multiple common LDL-C-increasing variants may resemble monogenic hypercholesterolemia (HCL) [30]. Genetic risk scores are used to evaluate the presence of polygenic HCL [31]. Studies have demonstrated that polygenic HCL is associated with an increased risk of premature atherosclerotic CVD [32]. A high polygenic risk of HCL was significantly associated with an increased risk of myocardial infarction (MI), ischemic stroke, coronary, and carotid revascularization [32].

Overall, the difficulties in diagnosing and differentiating genetically-based hyperlipidemias may stem from their wide phenotypic variability. Despite the availability of next-generation sequencing (NGS) and highly effective lipid-lowering therapy [1], genetically-based hyperlipidemias remain largely underdiagnosed and undertreated worldwide [13, 20, 33–35]. Investigation of clinical and biochemical features can play an important role in the identification and differentiation of atherogenic hyperlipidemias, the determination of disease prognosis, and the timely initiation of appropriate intensity treatment. A simultaneous comparison of clinical and biochemical features of several atherogenic hyperlipidemias such as FD, FH, polygenic, and severe HCL, has not been previously performed.

This study aimed to compare lipid levels, frequency, age at onset of coronary heart disease (CHD), and the severity of carotid and femoral atherosclerosis among patients with genetically-based hyperlipidemias (FH, FD, polygenic HCL), severe HCL, and those without lipid metabolism disorders.

## Materials and methods

### Sampling

Patients from three samples who had targeted (n = 3202) or exome (n = 172) sequencing data were analyzed for this study:

- The ESSE-Ivanovo sample consisted of subjects from the Ivanovo region (median age was 48 years old (37; 56); 36.5% were men, n = 1858; S1 Table), selected from the "Epidemiology of Cardiovascular Diseases and Risk Factors in Regions of the Russian Federation" (ESSE-RF) study, a cross-sectional study conducted across 13 regions of Russia from 2012 to 2013 [36];

- The ESSE-FH-RF sample (n = 88) included participants from the ESSE-FH-RF study who had a clinical diagnosis of definite or probable heterozygous FH. The ESSE-FH-RF study was a cross-sectional, non-interventional, and multicenter study. Participants were selected from the ESSE-RF study [4];

- The Russian patient sample (RPS) comprised of observed patients with diverse medical conditions, including lipid metabolism disorders, whose blood samples were collected at the Biobank of the National Medical Research Center (NMRC) for Therapy and Preventive Medicine (Moscow, Russia) (n = 1428) [37].

Inclusion criteria were age 18 years and older, targeted or exome sequencing data, available data or the ability to undergo ultrasound of the carotid and femoral arteries, and one of the following diagnoses:

- FD based on the ε2ε2 haplotype of the *APOE* gene and triglyceride (TG) levels ≥ 1.5 mmol/L, as described in a previous study [13] (n = 29);

- FH was diagnosed using the Dutch Lipid Clinic Network criteria [38]. The study enrolled patients with identified pathogenic or likely pathogenic variants in one of the following genes: *LDLR*, *APOB*, or *PCSK9*, and a "definite" FH diagnosis (≥ 9 points) (n = 61). As a result, participants in the study had both a positive genetic test result for FH and met the clinical criteria for FH;

- Polygenic HCL is characterized by elevated LDL- levels (> 4.9 mmol/L), a high polygenic risk score (PRS) for LDL-C (> 80th percentile), and the absence of tendon xanthomas (n = 49);

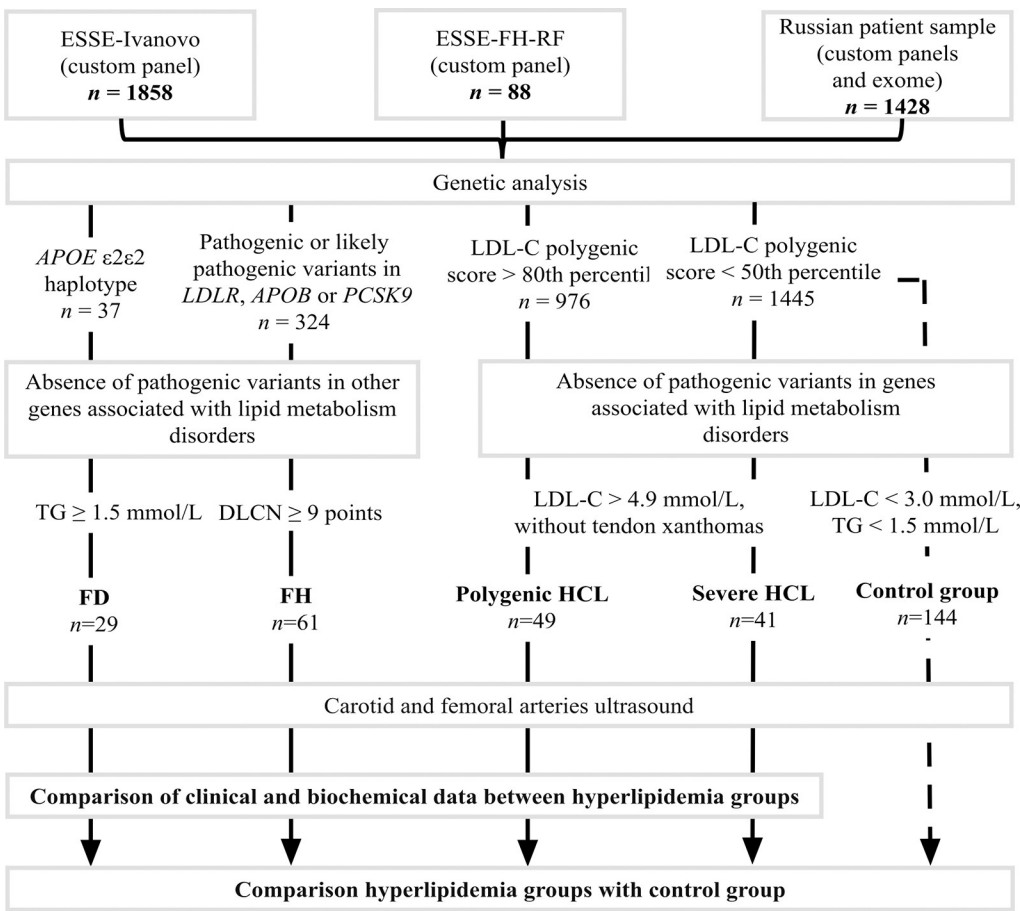

**Fig 1. Study design.** DLCN, Dutch Lipid Clinic Network criteria; FD, familial dysbetalipoproteinemia; FH, familial hypercholesterolemia; HCL, hypercholesterolemia; LDL-C, low-density lipoprotein cholesterol; TG, triglycerides.

- Severe HCL is defined by elevated LDL- levels (> 4.9 mmol/L), a low PRS for LDL-C (< 50th percentile), and the absence of tendon xanthomas (n = 41);

- The control group consisted of subjects without lipid disorders from the population-based sample (ESSE-Ivanovo). These subjects had no causal variants in 25 genes associated with lipid disorders included in the custom panel analyzed (see NGS section in Materials and methods) and had a low PRS for LDL-C (< 50th percentile). Their LDL-C levels were < 3.0 mmol/L and TG levels were < 1.5 mmol/L (n = 144).

    Exclusion criteria included pathogenic or likely pathogenic variants in genes associated with monogenic lipid metabolism disorders that were part of custom panels, except for *LDLR*, *APOB*, *PCSK9*, and *APOE*, as well as a combination of pathogenic or likely pathogenic variants in the *LDLR*, *APOB*, *PCSK9*, and *APOE* genes (Fig 1).

## Clinical and biochemical data

Retrospective clinical data from the ESSE-Ivanovo, ESSE-FH-RF, and RPS were used in the current study, including age, sex, body mass index (BMI), smoking status, the presence of hypertension and diabetes, and the presence and age at onset of CHD (including MI and coronary revascularization). The history of CHD was based on the medical records, and the

diagnosis was made according to current European clinical guidelines. Age at onset of CHD was defined as age at diagnosis. We also considered the presence of tendon xanthomas and the type, volume, and duration of lipid-lowering therapy (taking statins for more than 1 month before arterial ultrasound or for 3 months or longer historically).

Lipid levels, including LDL-C, high-density lipoprotein cholesterol (HDL-C), and TG, were measured using the Abbott Architect C-8000 system (Abbott Laboratories, North Chicago, IL, USA) and reported in mmol/L. Lipoprotein(a) (Lp(a)) levels were assessed with the Tokyo Boeki Sapphire-400 analyzer (Tokyo, Japan) and reported in mg/dL.

In the ESSE-Ivanovo and ESSE-FH-RF studies, LDL-C levels were determined directly. In this current study, LDL-C was measured directly for RPS participants with TG levels > 4.5 mmol/L. Otherwise, LDL-C levels were calculated using the Friedewald formula. For patients on regular statin therapy (37 patients, including ten with FD, eleven with FH, eight with severe HCL, and eight without lipid metabolism disorders who did not receive combined lipid-lowering therapy), the statin dose was converted to an equivalent atorvastatin dose. Pretreatment LDL-C levels were estimated using the average relative decrease in concentration with the appropriate atorvastatin dose [39]. HDL-C and TG levels were not recalculated in this case. Total cholesterol level was not included in the analysis due to the aims of this study. Lp(a) levels were reported without lipid-lowering therapy.

## Carotid and femoral arteries ultrasound

High-resolution B-mode ultrasonography was performed with a 12–5 MHz linear probe (Samsung Medison MySono U6) or a 17–5 MHz linear probe (Philips iU22 ultrasound system) (n = 321).

Patients from ESSE-Ivanovo underwent ultrasound 2–4 years after enrollment in ESSE-Ivanovo as part of the ATEROGEN-Ivanovo study [40]. In total, the ultrasound data of 258 participants of the ATHEROGEN-Ivanovo and ESSE-FH-RF studies [4] were used retrospectively. In addition, 60 patients from the RPS cohort underwent arterial ultrasound as part of the current study. In all these studies, ultrasound was performed according to the same protocol.

In the ESSE-RF study, the accuracy of the measurements was estimated by a qualified sonographer who also performed ultrasound examinations in patients from the ESSE-FH-RF and RPS cohorts. Retrospective data from three patients with FD from the RPS cohort were also included based on ultrasound protocols from the NMRC for Therapy and Preventive Medicine (plaque number, maximum stenosis, and total stenosis). Ultrasound data were not available for three patients with FD. However, due to the rare frequency of this disease, these patients were included in the overall study analysis.

All measurements were performed in both common carotid arteries (CCA) and available for ultrasound visualization of proximal segments of the internal carotid arteries (ICA). Both common femoral arteries (CFA) and 1.5 cm of the proximal segments of the superficial femoral arteries (SFA) were also examined. Plaque was defined as a focal structure that encroaches into the arterial lumen of at least 0.5 mm or 50% of the surrounding intima-media thickness (IMT) value or demonstrates a thickness ≥ 1.5 mm measured from the media-adventitia interface to the intima-lumen interface [41]. All measurements were made in diastole, corresponding to the R-wave of the electrocardiogram.

Carotid and femoral atherosclerosis was assessed using the following parameters: plaque number, maximum stenosis, total stenosis, maximum plaque height, and plaque score.

The presence of plaques was estimated at six sites of the carotid (femoral) arteries: the total length of both CCAs (CFAs), both carotid (femoral) bifurcations, and both proximal segments of ICAs (SFAs). Plaque number was defined as the sum of all plaques.

Percent diameter stenosis was defined at the site of maximum plaque obstruction in the artery in the transverse view and was obtained from measurements of the residual lumen area and the original area. It was calculated as a percentage according to the European Carotid Surgery Trial [42]. The maximum value of all percent area stenoses (maximum stenosis) and the sum of all maximum stenoses (total stenosis) obtained during carotid (femoral) arteries ultrasound were taken into account.

The plaque height was measured in cross-sectional view from the side in which a plaque was detected using a caliper placed at the adventitial plane and extending into the center of the lumen at right angles to the vessel wall. The maximum value of all detected plaque heights (maximum plaque height) was used for analysis.

For plaque score assessment, each carotid (femoral) artery was divided into four segments: 15 mm of proximal ICAs (SFAs) after the tip of the flow divider, the carotid bulb, and two distal parts of the CCAs (CFAs), each 15 mm. The plaque score was calculated by summing the maximum thickness of all plaques measured in millimeters on the near and far walls at each of the four parts of both sides of the carotid (femoral) arteries [43].

## Genetic analysis

**DNA extraction.** The blood samples were stored at −32˚C at the Biobank of the NMRC for Therapy and Preventive Medicine [37]. Genomic DNA was extracted from peripheral blood using the QIAamp DNA Blood Mini Kit (Qiagen, Hilden, Germany). DNA concentration was measured using the Qubit 4 Fluorimeter (Thermo Fisher Scientific, Waltham, MA, USA).

**NGS.** NGS was performed on a Nextseq 550 (Illumina, San Diego, CA, USA), resulting in paired-end reads (150 or 300 bp). For targeted sequencing (custom panels), the libraries were prepared with the SeqCap EZ Prime Choice Library kit (Roche, Basel, Switzerland). Exome libraries were prepared with the IDT-Illumina TruSeq DNA Exome protocol (Illumina, San Diego, CA, USA). All stages of sequencing were conducted according to the manufacturers' protocols.

A set of 24 genes associated with lipid metabolism disorders (*ABCA1*, *ABCG5*, *ABCG8*, *ANGPTL3*, *APOA1*, *APOA5*, *APOB*, *APOC2*, *APOC3*, *APOE*, *CETP*, *GPD1*, *GPIHBP1*, *LCAT*, *LDLR*, *LDLRAP1*, *LIPC*, *LIPI*, *LMF1*, *LPL*, *PCSK9*, *SAR1B*, *STAP1*, *USF1*) was analyzed in exome sequencing data and one of the custom panels [13]. The other custom panel included 6 genes associated with lipid metabolism disorders (*ANGPTL3*, *APOB*, *APOC3*, *LDLR*, *LPL*, *PCSK9*) and rs7412, rs429358 of the *APOE* gene (Fig 2).

**Bioinformatic analysis and clinical interpretation.** The GRCh37/hg19 reference genome was selected for aligning paired-end reads. A custom-designed pipeline [44], based on GATK 3.8 [45], was used for processing sequencing data and quality control. Individual variant call format files were then generated for each sample, containing a list of variants, their genomic coordinates, coverage data, and other characteristics. Low-quality variants, probably due to sequencing errors, were filtered out. The coverage depth of the reference and alternative alleles, read and mapping quality, and other relevant factors were reported and analyzed.

Variants with a minor allele frequency < 0.01% or those that were missing in the Genome Aggregation Database (gnomAD; http://gnomad.broadinstitute.org) and were non-synonymous were selected. Variant annotation was performed using Online Mendelian Inheritance in Man database [46], gnomAD (v2.1.1) [47], ClinVar [48], Human Gene Mutation Database [49], Leiden Open Variation Database [50], dbSNP [51], literature data, and segregation information. Clinical interpretation was based on the American College of Medical Genetics and Genomics/Association for Molecular Pathology guidelines [52] and the Clinical Genome Resource guidelines for *LDLR* variant classification [53].

                                     

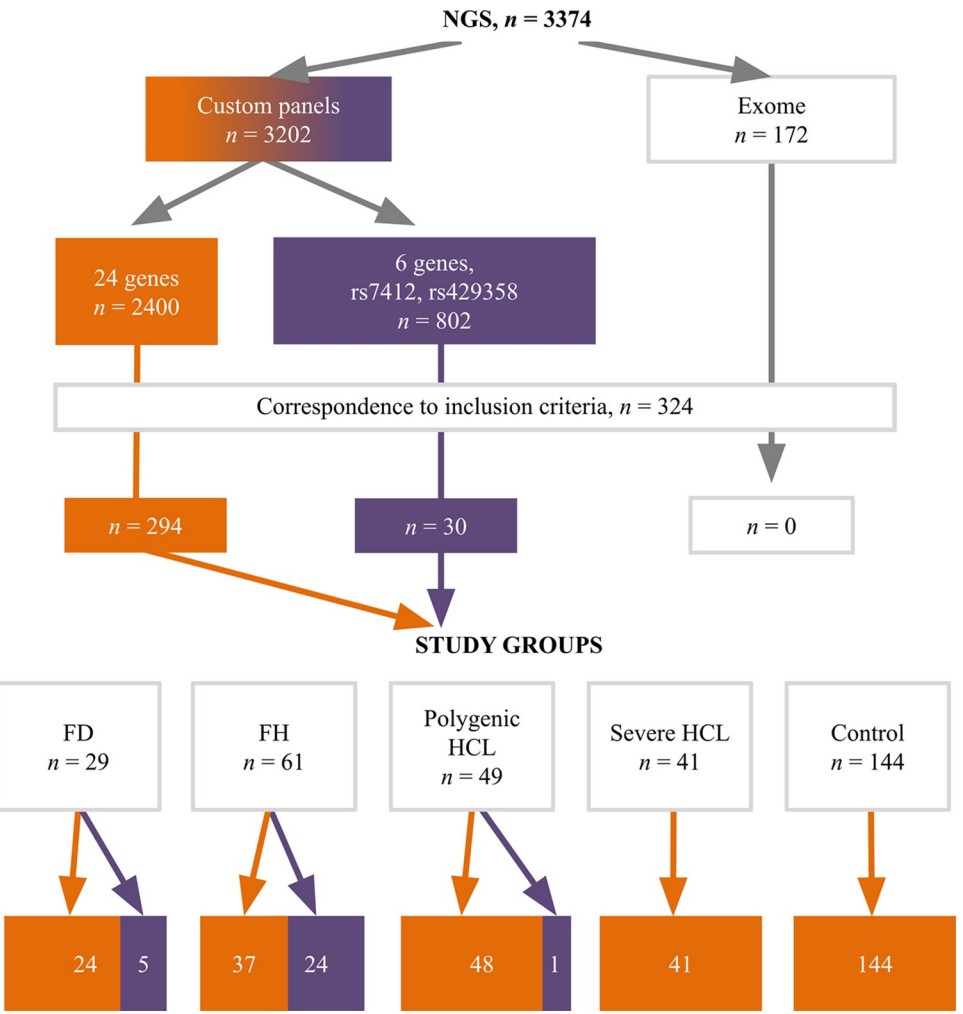

**Fig 2. Distribution of analyzed patients by NGS data.** The orange color indicates a 24-gene custom panel, while the purple color indicates a 6-gene custom panel. FD, familial dysbetalipoproteinemia; FH, familial hypercholesterolemia; HCL, hypercholesterolemia; NGS, next-generation sequencing.

The ε2ε2 haplotype of the *APOE* gene was identified as homozygous for the ε2 allele (rs7412) with no alteration in rs429358, as previously described [13].

Sanger sequencing was performed using the DNA sequencer Applied Biosystem 3500 Genetic Analyzer (Thermo Fisher Scientific, Waltham, MA, USA) in conjunction with the ABI PRISM BigDye Terminator v3.1 reagent kit (Thermo Fisher Scientific, Waltham, MA, USA), following the manufacturer's protocol.

**PRS.** The weighted PRS was calculated using the b-coefficients from original articles that previously demonstrated significant associations with LDL-C (57 variants predicting LDL-C levels [31]) or TG (40 variants predicting TG levels [54]) in the population of the European part of Russia. PRS of the study participants was compared with that of all 1858 subjects from the population-based cohort (ESSE-Ivanovo). The *APOE* haplotype was included to determine the final genetic score for LDL-C levels: -0.9 for ε2ε2, -0.4 for ε2ε3, -0.2 for ε2ε4, 0 for ε3ε3, 0.1 for ε3ε4, and 0.2 for ε4ε4 [55]. A high PRS of HCL or hypertriglyceridemia was defined as a weighted PRS > 80th percentile, whereas a low PRS was indicated by a weighted PRS < 50th percentile. For LDL-C, a high polygenic contribution was represented by a range of weighted

PRS from (-2.054) to (-2.937) in the ESSE-Ivanovo sample. Values from (-3.215) to (-4.416) were set for a low contribution (S1A Fig). Similarly, a high polygenic contribution to TG levels was characterized by a range of weighted PRS from (-1.213) to (-1.841), whereas a low contribution was in the range from (-2.037) to (-2.820) (S1B Fig). PRS for LDL-C and TG was calculated for all study participants, except three patients with FH.

## Ethical statement

The study was conducted in accordance with the Declaration of Helsinki and the National Standard of the Russian Federation "Good Clinical Practice (GCP)" GOST R52379-2005 and was approved by the Independent Ethics Committee of NMRC for Therapy and Preventive Medicine (protocol number 07-05/20 dated November 26, 2020). In order to comply with the above-mentioned laws, as well as Article 93 of the Federal Law "On the Fundamentals of Health Protection of Citizens of the Russian Federation" dated November 21, 2011, No. 323-FZ, each subject signs a written consent to the processing of their personal data. Data from the ESSE-Ivanovo study, including the ATEROGEN-Ivanovo study, the ESSE-FH-RF study, and the RPS cohort, were used in the current study. Written informed consent was obtained from each patient as part of their participation in these scientific projects. In addition, written informed consent was obtained from 60 patients in the RPS cohort who underwent arterial ultrasound as part of the current study. ESSE-Ivanovo, ESSE-FH-RF and RPS data have been available since November 27, 2020. The database containing clinical, biochemical, and genetic data was de-identified and encrypted to ensure confidentiality.

## Statistical methods

Statistical analyses were performed using R version 4.3.2 (R Foundation for Statistical Computing, Vienna, Austria) [56]. Continuous variables were summarized as median (Me) and interquartile range (Q1; Q3), while categorical variables were presented as absolute numbers and percentages. The Mann-Whitney U test was used to compare continuous variables between two independent groups, and the two-sided Fisher's exact test was used for categorical variables. For comparisons between three or more independent groups, the Kruskal-Wallis test was utilized for continuous variables and the two-sided Fisher's exact test for categorical variables. The p-values from pairwise comparisons were adjusted using the Holm-Bonferroni method.

Comparison of lipid levels between groups with adjustments for sex and age was performed using logistic regression. The Mann-Whitney test was used for Lp(a). Comparison of the frequency of CHD and the severity of carotid and femoral atherosclerosis between hyperlipidemia groups with adjustments for sex, age, BMI, arterial hypertension, diabetes, smoking, and statin treatment duration was performed using logistic regression. For this purpose, continuous parameters were discretized relative to the sample median. Subgroups were included in the model as dummy variables, and the categorical or discretized continuous parameter was included as a target variable. The p-values of the described regression pairwise comparisons were adjusted using the Holm-Bonferroni method. In the control group, a single-factor analysis was performed due to the low proportion of subjects with CHD and the presence of carotid and femoral atherosclerosis. When the low proportion of subjects with CHD in the polygenic and severe hyperlipidemia HCL groups prevented the use of logistic regression, the Mann-Whitney test was used. A p-value of less than 0.05 was considered statistically significant. Data visualization was carried out using the ggplot2 package [57] and the Viridis color palette [58].

**Statistical power analysis.** A statistical power analysis was conducted to assess the study's expected detectable effect size, using data from the ESSE-Ivanovo study. The sample sizes and

standard deviation of the parameter under investigation from the ESSE-Ivanovo data were considered for this analysis. The probability of a Type I error (false positive result) was set at a two-sided significance level of $\alpha = 0.05$, while the Type II error (false negative result) probability was set at $\beta = 0.2$. The analysis was performed using the non-central Student T-distribution with the assistance of an online calculator [59]. We suppose that the effect size values obtained in this study are reasonable and expected (Table 1).

## Results

### Clinical characteristics of the participants

Overall, 324 patients were included in this study (Table 2).

Among patients with hyperlipidemia (FD, FH, polygenic, and severe HCL), the median age was 54 years (47; 61), and 40.0% were men. More than half (75.8%) had a BMI > 24.9 kg/m$^2$, and 39.9% were obese. The number of ever smokers was 2.4 times higher in men than in women (p < 0.001). In addition, 47.2% of patients were taking statins (Table 2).

Patients with FD and FH were younger compared to those with severe or polygenic HCL. Patients with FD had a higher BMI in comparison to patients with FH, but comparable to patients with polygenic or severe HCL. Patients with polygenic HCL were more likely to have hypertension. The percentage of subjects taking statins was highest in patients with FH (83.6%), which was significantly different from the other study groups (p < 0.001) (S2 Table).

Patients with FH were more likely to have polygenic HCL than those with FD (25.4% vs. 0%, p = 0.002). On the other hand, there were no significant differences in LDL-C levels between FH patients with a high PRS for LDL-C levels and those with a PRS ≤ 80th percentile (p = 0.620). Patients with FD were more likely to have polygenic hypertriglyceridemia than those with severe HCL (51.7% vs. 12.2%, p = 0.003). However, TG levels were similar between patients with a high PRS for TG and those with a PRS ≤ 80th percentile in both the FD (p = 0.156) and severe HCL (p = 0.780) groups (Table 2).

The median age of subjects in the control group was 52 years (45; 59) (p = 0.003 and p = 0.002 compared to patients with polygenic and severe HCL, respectively), and 27.1% of them were men (p = 0.029 and p = 0.014 compared to patients with FD and FH, respectively). Additionally, 4.2% of subjects were taking statins (Table 2).

### Lipid levels

Overview of study findings presented in Fig 3.

Comparison of LDL-C, HDL-C, TG, and Lp(a) levels among genetically-based hyperlipidemia groups, severe HCL, and compared to the control group is shown in Fig 4.

**Table 1. Statistical power analysis.**

| Parameter[a] | Expected detectable effect size for carotid arteries parameters | Expected detectable effect size for femoral arteries parameters |
|---|---|---|
| Maximum stenosis, % | 11.2 | 10.4 |
| Total stenosis, % | 34.8 | 32.4 |
| Maximum plaque height, mm | 3.2 | 4.6 |
| Plaque score, mm | 2.3 | 3.1 |

[a]For samples: FD (n = 29) and severe HCL (n = 41). Statistical power 80.0%.

**Table 2. Clinical characteristics of the participants.**

| Parameter | All[a] (n = 180) | FD (n = 29) | FH (n = 61) | Polygenic HCL (n = 49) | Severe HCL (n = 41) | p-value[b] | Control group (n = 144) | p-value[c] |
|---|---|---|---|---|---|---|---|---|
| Men, n (%) | 72 (40.0) | 14 (48.3) | 28 (45.9) | 16 (32.7) | 14 (34.1) | 0.341 | 39 (27.1) | **0.018** |
| Age, years, Me (Q1; Q3) | 54 (47; 61) | 50 (46; 59) | 50 (40; 61) | 56 (52; 60) | 58 (53; 61) | **0.027** | 52 (45; 59) | **0.044** |
| BMI, kg/m², Me (Q1; Q3) | 28.8 (25.2; 32.7) | 29.0 (26.8; 32.5) | 26.3 (23.6; 30.0) | 29.5 (27.3; 33.7) | 29.6 (27.4; 32.6) | **0.002** | 27.3 (24.0; 30.9) | 0.433 |
| | n = 178 | | | n = 48 | n = 40 | | | |
| Current smoking, n (%) | 29 (16.1) | 9 (31.0) | 9 (14.8) | 7 (14.3) | 4 (9.8) | 0.132 | 19 (13.2) | 0.638 |
| Ex-smokers, n (%) | 28 (15.6) | 6 (20.7) | 12 (19.7) | 6 (12.2) | 4 (9.8) | 0.430 | 22 (15.3) | 1.0 |
| Hypertension, n (%) | 128 (71.1) | 20 (69.0) | 28 (45.9) | 39 (79.6) | 41 (100) | **< 0.001** | 103 (71.5) | 1.0 |
| Diabetes, n (%) | 15 (8.3) | 2 (6.9) | 4 (6.6) | 5 (10.2) | 4 (9.8) | 0.914 | 9 (6.3) | 0.528 |
| Statins, n (%) | 85 (47.2) | 13 (44.8) | 51 (83.6) | 11 (22.4) | 10 (24.4) | **< 0.001** | 6 (4.2) | **< 0.001** |
| High LDL-C PRS, n (%) | 64 (36.2) | 0 | 15 (25.9) | 49 (100)[d] | 0[d] | **0.002**[e] | 0[d] | NA[f] |
| | n = 177 | | n = 58 | | | | | |
| High TG PRS, n (%) | 53 (29.9) | 15 (51.7) | 17 (29.3) | 16 (32.7) | 5 (12.2) | **0.004** | 13 (9.0) | **< 0.001** |
| | n = 177 | | n = 58 | | | | | |

[a]Data includes FD, FH, polygenic and severe HCL.

[b]p-value indicates differences among FD, FH, polygenic, and severe HCL. The Kruskal–Wallis test was used for continuous variables and the two-tailed Fisher's exact test for categorical variables.

[c]p-value indicates differences among the control group and FD, FH, polygenic, and severe HCL. The Kruskal–Wallis test was used for continuous variables and the two-tailed Fisher's exact test for categorical variables.

[d]Inclusion criteria.

[e]p-value indicates differences between FD and FH, obtained by pairwise comparisons using the two-tailed Fisher's exact test.

[f]Not applicable.

After adjustment for sex and age, LDL-C levels were significantly higher in patients with FH. Patients with polygenic HCL had higher LDL-C levels compared to patients with severe HCL (p = 0.046) (Fig 4A).

LDL-C (median 3.57 mmol/L) and HDL-C (median 1.03 mmol/L) levels were lower in patients with FD compared to those with FH, polygenic, and severe HCL, after adjustment for sex and age. HDL-C levels in men and women with FD were 0.98 (0.79; 1.05) mmol/L and 1.12 (0.83; 1.47) mmol/L, respectively, and were significantly lower than those in the control group (p < 0.001) (Fig 4A and 4B). Patients with FD had significantly higher TG levels (median 4.10 mmol/L) compared to those with FH, polygenic, and severe HCL after adjustment for sex and age. The distribution of TG levels ranged from a minimum of 1.54 mmol/L to a maximum value of 13.12 mmol/L (Fig 4C).

Patients with genetically-based hyperlipidemias (FD, FH, and polygenic HCL) exhibited significantly lower HDL-C levels compared to subjects without lipid metabolism disorders (Fig 4B).

Lp(a) levels were available for 141 of 180 patients with hyperlipidemia and for all subjects without lipid metabolism disorders (control group, n = 144). There were no differences in Lp (a) levels among patients with genetically-based hyperlipidemias and severe HCL (p = 0.202). However, patients with severe HCL had higher Lp(a) levels compared to the control group (p < 0.001) (Fig 4D). Specifically, Lp(a) levels among genetically-based hyperlipidemias were distributed as follows: 71.0% had levels < 30.0 mg/dL, 10.0% between 30.0–49.0 mg/dL, and 20.0% had levels ≥ 50.0 mg/dL. Only one patient (in the FH group) had levels > 180.0 mg/dL. The distribution of Lp(a) levels in all groups is presented in Fig 5.

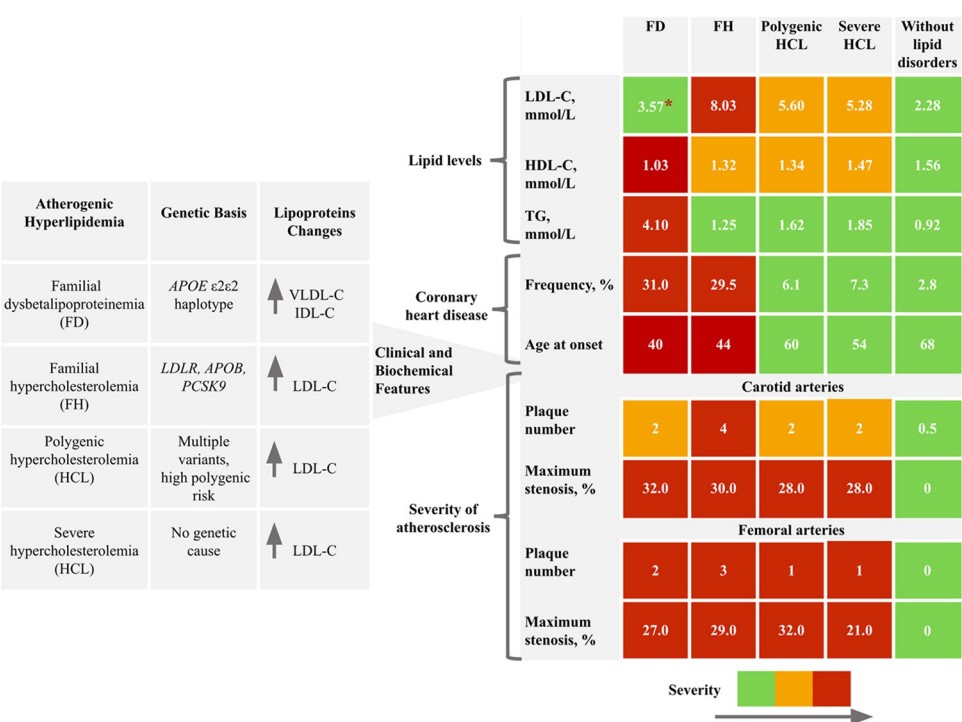

**Fig 3. Overview of study findings.** The summary results table uses color coding to indicate the severity of parameter values: green represents the significantly lower severity. Red represents the significantly higher severity. When differences were found between several groups, we used orange to indicate intermediate severity between green and red. Only the most significant ultrasound parameters of carotid and femoral atherosclerosis (plaque number and maximum stenosis) are presented. *Median values are shown in all cells. IDL-C, intermediate-density lipoprotein cholesterol; VLDL-C, very low-density lipoprotein cholesterol.

## Frequency and age at onset of CHD

Comparison of the frequency and age at onset of CHD (the age of diagnosis) among genetically-based hyperlipidemia groups, severe HCL, and comparison with the control group are summarized in Fig 6.

Patients with FH and FD had an early onset of CHD (the median age was 44 and 40 years, respectively) (Fig 6A). Regardless of sex, age, BMI, arterial hypertension, diabetes, smoking, and duration of statin treatment, a comparable frequency of CHD was observed in both groups (29.5% and 31.0%, respectively, p = 1.0). The frequency of CHD, based on all adjustments, was significantly higher in patients with FD compared to those with polygenic HCL (Fig 6B). In patients with polygenic and severe HCL, the proportion of subjects with CHD was small: 6.1% and 7.3%, respectively, and comparable (p = 1.0) (Fig 6B).

In the control group, there was an extremely small proportion of subjects with CHD (2.8%), including coronary artery revascularization, and none had a history of MI. The frequency of CHD in patients with polygenic (6.1%) and severe HCL (7.3%) was comparable to that in the control group (p = 0.373 and p = 0.368, respectively). At the same time, patients with FH and FD had a significantly higher frequency of CHD compared to subjects without lipid metabolism disorders (Fig 6B).

## Severity of carotid and femoral atherosclerosis

We analyzed ultrasound parameters of carotid and femoral atherosclerosis among genetically-based hyperlipidemia groups, severe HCL, and compared them with the control group. The

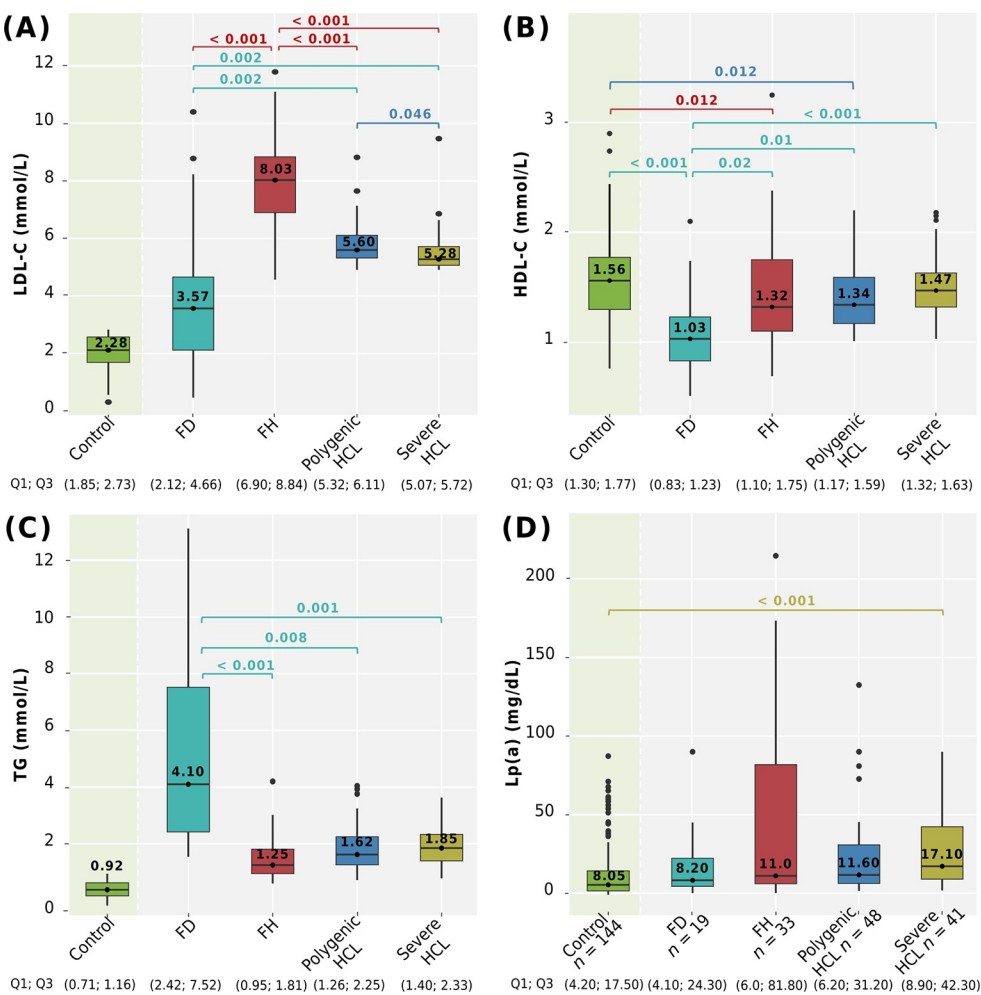

**Fig 4. Lipid levels among hyperlipidemia groups and compared with the control group.** A: LDL-C; B: HDL-C; C: TG; D: Lp(a) levels. entral lines represent the median, box limits represent upper and lower quartiles, vertical lines represent 1.5 times the quartile range, and points represent outliers. Horizontal square brackets with p-values above indicate significantly different groups. Only significant differences are indicated among groups. P-values were adjusted by the Holm-Bonferroni method. FD, familial dysbetalipoproteinemia; FH, familial hypercholesterolemia; HCL, hypercholesterolemia; HDL-C, high-density lipoprotein cholesterol; LDL-C, low-density lipoprotein cholesterol; Lp(a), lipoprotein(a); TG, triglycerides.

main differences between all groups are summarized in Fig 7, showing plaque number, total stenosis, and plaque score. A pairwise comparison of maximum stenosis and maximum plaque height is presented in S3 Table.

After all adjustments, it was found that patients with FH had a significantly higher carotid plaque number (median 4, Fig 7A), a greater severity of total carotid stenosis (median 106%, Fig 7B), and a higher carotid plaque score (median 5.93 mm, Fig 7C) compared to those with FD, polygenic HCL, and severe HCL. The severity of carotid atherosclerosis among patients with FD, polygenic HCL, or severe HCL was comparable after all adjustments (Fig 7A–7C, and S3 Table). Ultrasound parameters of femoral atherosclerosis among patients with FH, FD, polygenic HCL, and severe HCL showed no significant differences (Fig 7D–7F and S3 Table).

Patients with FD, FH, polygenic HCL, or severe HCL had more pronounced carotid and femoral atherosclerosis compared to subjects without lipid metabolism disorders (p < 0.001, Fig 7 and S3 Table).

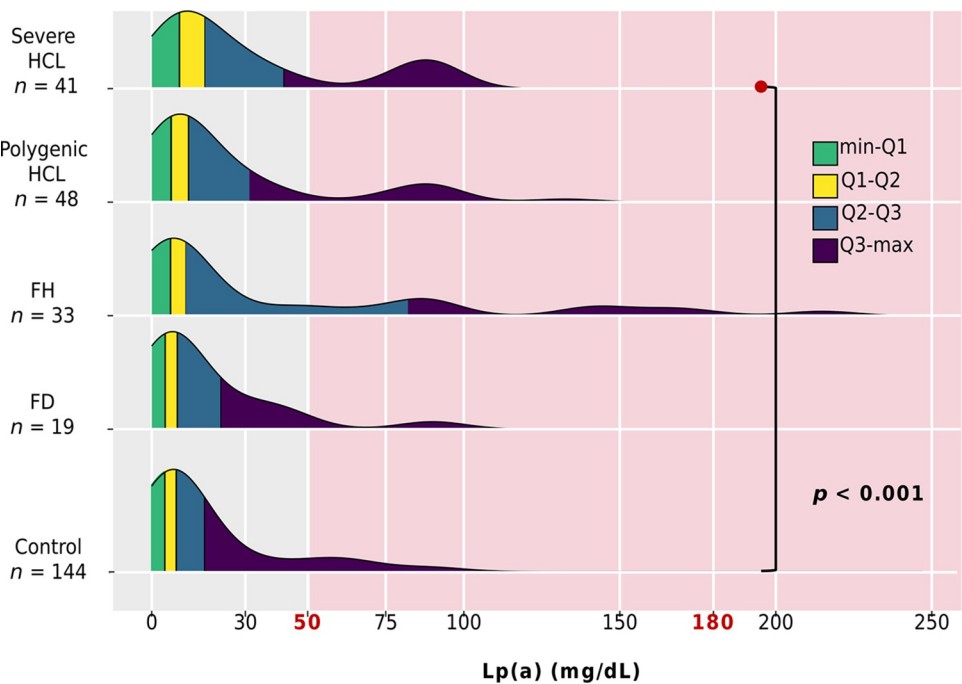

**Fig 5. Distribution of Lp(a) by group.** Ranges: minimum-Q1 highlighted in green, Q1-Q2 in yellow, Q2-Q3 in blue, and Q3-maximum in purple. The red background indicates an elevated Lp(a) level ($\geq$ 50.0 mg/dL). Threshold >180 mg/dL indicates very high Lp(a) level. Only significant differences are indicated. FD, familial dysbetalipoproteinemia; FH, familial hypercholesterolemia; HCL, hypercholesterolemia; Lp(a), lipoprotein(a).

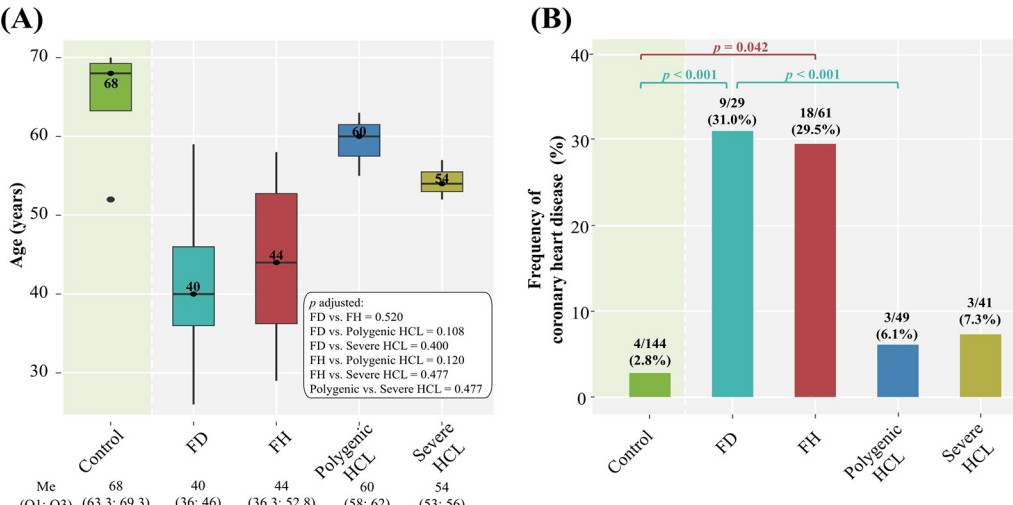

**Fig 6. Frequency and age at onset of CHD among hyperlipidemia groups compared with the control group.** A: age at onset. Central lines represent the median, box limits represent upper and lower quartiles, vertical lines represent 1.5 times the quartile range, and points represent outliers. Only significant differences are indicated among groups. P-values were adjusted by the Holm-Bonferroni method; B: frequency. Horizontal square brackets with p-values above indicate significantly different groups. Only significant differences are indicated among groups. P-values were adjusted by the Holm-Bonferroni method. FD, familial dysbetalipoproteinemia; FH, familial hypercholesterolemia; HCL, hypercholesterolemia; Me, median.

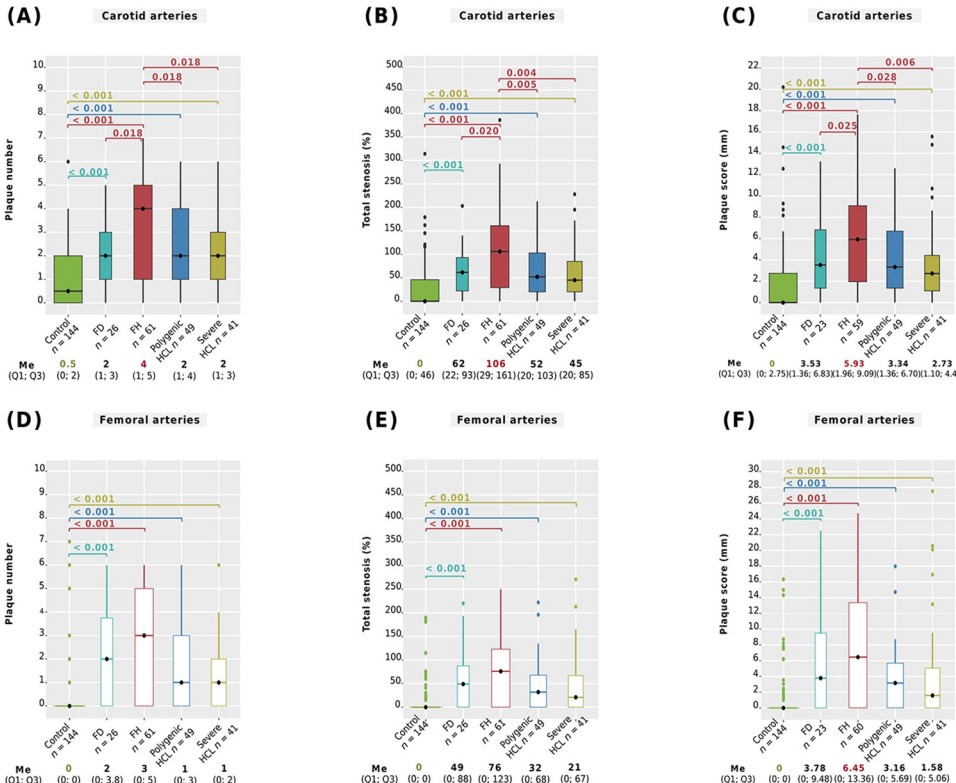

**Fig 7. Severity of carotid and femoral atherosclerosis.** A: number of carotid plaques; B: total carotid stenosis; :
carotid plaque score; D: number of femoral plaques; E: total femoral stenosis; F: femoral plaque score. Central lines
represent the median, box limits represent upper and lower quartiles, vertical lines represent 1.5 times the quartile
range, and points represent outliers. Horizontal square brackets with p-values above indicate significantly different
groups. Only significant differences are indicated among groups. For study groups, p-values were obtained from linear
regression with adjustments, and from the Mann-Whitney test for control vs. study groups. P-values were adjusted by
the Holm-Bonferroni method. FD, familial dysbetalipoproteinemia; FH, familial hypercholesterolemia; HCL,
hypercholesterolemia; Me, median.

## Discussion

### Lipid levels

A phenotypic feature of FH is the presence of extremely elevated lifelong LDL-C levels [1]. In
our study, among the analyzed hyperlipidemia groups, the highest pretreatment LDL-C levels
(median 8.03 mmol/L) were found specifically in patients with FH. Several studies have
reported similar findings. Trinder et al., 2019 showed that patients with FH had higher mean
LDL-C levels than subjects with nongenetic HCL (7.08 mmol/L vs. 5.38 mmol/L, p < 0.001)
[60]. D'Erasmo et al., 2021 [61] and Mickiewicz et al., 2020 [62] demonstrated the highest
LDL-C levels in patients with FH compared to patients with polygenic HCL (median 6.72
mmol/L vs. 5.56 mmol/L, p < 0.001 and mean 7.6 mmol/L vs. 6.2 mmol/L, p < 0.001). How-
ever, opposite results are also available. Trinder et al., 2020 found that median LDL-C levels
after adjustment for lipid-lowering therapy did not differ between monogenic (4.17 mmol/L),
polygenic (4.03 mmol/L), and severe nongenetic (4.02 mmol/L) HCL [32]. In our study,
LDL-C levels in patients with polygenic HCL were significantly higher than in patients with
severe HCL. This concurs with D'Erasmo et al., 2021 findings [61], where mean LDL-C levels
were higher in patients with polygenic HCL compared to nongenetic HCL (5.05 mmol/L,
p = 0.014). The difference in results between various studies may be due to the different

definitions of polygenic HCL. This may involve the different number of genetic variants in the PRS and the stratification of polygenic risk. In our study, polygenic HCL was defined as a genetic score > 80th percentile based on 57 variants, similar to that of Trinder et al., 2019 [60]. D'Erasmo et al., 2021 achieved results comparable to ours using six variants and PRS above the 70th percentile [61]. In the study by Trinder et al., 2020, 223 variants were included, and a high PRS was defined as an LDL-C level > 95th percentile [32]. A strength of our study is the assessment of PRS based on a population sample.

In addition, we observed several phenotypic patterns in patients with FD. First, they had the lowest LDL-C levels of all the hyperlipidemia groups analyzed. Although patients with FD had the lowest LDL-C levels overall, the median value of 3.57 mmol/L exceeds the recommended level for patients at low cardiovascular risk [1]. This is the first study to compare LDL-C levels in FD versus FH, polygenic, and severe HCL. Second, the highest TG levels (median 4.10 mmol/L) were found specifically in patients with FD. A similar feature was highlighted by Bea et al., 2017 among patients with FD (median 3.60 mmol/L) versus FH (1.16 mmol/L) and polygenic HCL (1.28 mmol/L), p < 0.05 [63]. In our study, there was wide variability in TG levels among patients with FD, ranging up to 13.12 mmol/L, but the median was 4.10 mmol/L, indicating a moderate increase in TG levels. This is consistent with median TG values of 2.48 mmol/L [23], 2.85 mmol/L [16], and values observed in other studies [29, 64, 65]. Third, patients with FD had lower HDL-C levels (median 1.03 mmol/L) not only compared to patients with genetically-based hyperlipidemias or severe HCL, but also compared to subjects without lipid disorders. This concurs with findings from other studies, where median HDL-C levels were reported as 0.87 mmol/L [65] and 0.92 mmol/L [63]. One study reported a median HDL-C level of 1.3 mmol/L in 63 patients with FD, falling within the normal range [64].

The well-established epidemiological association between low HDL-C levels and an increased risk of atherosclerotic CVD has been demonstrated [66]. Our study revealed that HDL-C levels were lower in patients with genetically-based hyperlipidemias compared to the control group. While HDL-C levels are influenced by various metabolic, genetic, and epigenetic factors, they can also serve as an indicator of impaired LDL-C and TG metabolism in patients with genetically-based hyperlipidemias [66].

Patients with severe HCL exhibited elevated Lp(a) levels compared to the control group. However, the median level of 11.60 mg/dL observed in this group falls within the normal range and aligns with data from the general population. For instance, data from the UK Biobank revealed a median Lp(a) level of 19.0 nmol/L in 412,724 subjects, approximately equivalent to 8.1 mg/dL [11].

## Frequency and age at onset of CHD

Based on the results of the current study, patients with FH and FD are more likely to develop CHD compared to individuals without lipid disorders. In addition, we observed that patients with FH and FD have a comparable early onset and frequency of CHD. Thus, by the age of 44 years for FH and 40 years for FD, patients may already have CHD. Furthermore, the frequency of CHD was higher in patients with FD compared to those with polygenic HCL.

Several studies have demonstrated a causal relationship between elevated levels of cholesterol-enriched remnant lipoproteins in FD and an increased risk of atherosclerotic CVD, such as MI, coronary revascularization, and ischemic stroke [67–69]. However, the number of studies specifically investigating the association between FD and atherosclerotic CVD is limited. For instance, Hopkins et al., 2005 found an independent association between FD and the risk of premature HD. FD was defined using ultracentrifugation without genetic testing [70]. In a

large European study by Koopal et al., 2015, the prevalence of CHD in patients with FD was 19.0%, consistent with our result of 31.0%. In most cases, FD was identified based on the ε2ε2 haplotype of the *APOE* gene or the Arg136Ser variant. However, 20.0% of patients exhibited only an FD phenotype (determined by isoelectric focusing) and did not undergo genetic testing to confirm FD [25].

In the current study, the frequency of CHD in patients with polygenic HCL did not differ significantly from that in patients with severe HCL or subjects without lipid metabolism disorders. Initially, Trinder et al. (2019) found no significant differences in CVD risk between patients with polygenic and severe HCL (p = 0.30). However, there was a noticeable trend towards increased CVD risk, particularly in patients with polygenic HCL [60]. Upon expanding the patient cohort, the authors confirmed a higher risk of CVD in patients with polygenic HCL compared to those with severe HCL (p < 0.001) [32].

Therefore, our findings emphasize the impact of prolonged exposure to LDL-C from birth in individuals with FH, as well as the significant role of highly atherogenic cholesterol-enriched remnant lipoproteins in patients with FD, in the early onset of CHD. Early detection of FH and FD in individuals with hyperlipidemia is essential for preventing cardiovascular events.

## Severity of carotid and femoral atherosclerosis

We examined ultrasound parameters of carotid and femoral atherosclerosis, reflecting the spread of the atherosclerotic process both along the lumen (plaque number, total stenosis, and plaque score) and into the lumen of the arteries (maximum stenosis and maximum plaque height).

We observed a greater severity of carotid and femoral atherosclerosis among all examined hyperlipidemia groups compared to subjects without lipid disorders. Thus, elevated levels of LDL-C and cholesterol-enriched remnant lipoproteins contribute to the progression of carotid and femoral atherosclerosis, regardless of the molecular basis.

In addition, we identified prolonged atherosclerotic damage in the carotid arteries as a phenotypic feature of FH, indicating long-term exposure to LDL-C. In contrast, there were no significant differences in femoral atherosclerosis between patients with FH versus FD, polygenic HCL, or severe HCL.

Patients with FD, polygenic HCL, and severe HCL showed no significant differences in the severity of carotid and femoral atherosclerosis. It is plausible that an early onset of CHD is a potential feature of FD, where cholesterol-enriched remnant lipoproteins may have a greater impact on the development of coronary atherosclerosis compared with peripheral atherosclerosis.

A comprehensive comparison of ultrasound parameters of carotid and femoral atherosclerosis in patients with FH, FD, polygenic HCL, and severe HCL has not been previously performed. Bea et al., 2017 found a higher frequency of carotid plaque presence in patients with polygenic HCL (43.2%) compared to those with FH (31.9%), p < 0.05. The prevalence of carotid plaques was comparable between patients with FH and FD (45.5%), and between FD and polygenic HCL [63]. This is the only published study to evaluate carotid atherosclerosis in patients with FD or compare them to FH and polygenic HCL. In our current study, we found that carotid plaques were detected via ultrasound in 88.5% of patients with FH, 84.6% of patients with FD, and 83.7% of patients with polygenic HCL. Bea et al., 2017 only analyzed the presence of carotid plaque, which limits the comprehensive assessment of carotid atherosclerosis. In addition, the authors did not use polygenic scales to diagnose polygenic HCL, potentially impacting the study's results. Sharifi et al., 2017 measured the carotid IMT in patients

with FH (n = 56) and polygenic HCL (n = 30). The mean of all the carotid IMT measurements was significantly greater in FH than the polygenic patients (0.74 mm vs. 0.66 mm, p = 0.038) [71].

Thus, previous studies have focused on IMT or the presence of carotid plaque when comparing genetic hyperlipidemias. While the presence of carotid plaque in patients with FD has been evaluated in one study [63], it is noteworthy that ultrasound parameters of femoral atherosclerosis in FD patients have not been previously evaluated. In addition, carotid atherosclerosis in FH has been studied more extensively than femoral atherosclerosis. Given the early onset of CHD in patients with FH and FD, along with the valuable of ultrasound markers in estimating cardiovascular risk, it is important to evaluate the severity of carotid and femoral atherosclerosis in patients with atherogenic hyperlipidemias with different genetic basis. This is the first study to provide a comprehensive comparison of carotid and femoral atherosclerosis in patients with genetically-based hyperlipidemias, severe HCL, and subjects without lipid disorders.

## Limitations

This study has several limitations. First, lipoprotein ultracentrifugation or electrophoresis were not used to confirm the FD phenotype. Instead, the diagnosis of FD was based on the ε2ε2 haplotype of the *APOE* gene and TG levels ≥ 1.5 mmol/L. We have previously described this approach and its applicability [13]. Second, pretreatment TG and HDL-C levels were not recalculated for patients on regular statin therapy because of the lack of reliable data on the impact of different statin doses on these parameters. Third, 30 patients included in the study were analyzed using a custom panel of 6 genes associated with lipid metabolism disorders. Therefore, it is possible that these patients have pathogenic variants in other genes associated with lipid disorders. However, the selected genes in this panel are the most important for FH diagnosis. Of note, the majority of patients in the study, including all controls, those with severe HCL, and 48 of 49 patients with polygenic HCL, underwent sequencing of 24 lipid-associated genes. Lastly, the absence of significant differences in CHD frequency between patients with FD and severe HCL, or between patients with FH and those with polygenic HCL and severe HCL, is probably explained by the small sample size. Nevertheless, the statistical power was set at 80% for the main parameters investigated in this study, using smaller sample sizes as a reference.

## Conclusions

Through a comprehensive simultaneous comparison of genetically-based hyperlipidemias (FH, FD, polygenic HCL), severe HCL, and individuals without lipid disorders, this study highlights the most important distinct phenotypic features unique to each group. Patients with FH and FD have a similar early onset and frequency of CHD. They are also more likely to develop CHD and have a greater severity of carotid and femoral atherosclerosis compared to subjects without lipid disorders. Furthermore, prolonged atherosclerotic damage in the carotid arteries emerged as a phenotypic feature of FH. These findings underline the impact of long-term exposure to LDL-C and cholesterol-enriched remnant lipoproteins on the early onset of CHD and the progression of carotid and femoral atherosclerosis. Overall, the results of this study provide an opportunity to improve the differential diagnosis of atherogenic hyperlipidemias in daily clinical practice, contribute to the understanding of the phenotypic features of understudied FD, and underscore FH and FD as the most atherogenic hyperlipidemias.

## Supporting information

**S1 Fig. PRS in the ESSE-Ivanovo sample (n = 1858).** A: LDL-C levels; B: TG levels. The x-axis shows the distribution of PRS, and the y-axis shows the number of subjects. Color

indicates percentiles.
(TIF)

**S1 Table. Characteristics of the ESSE-Ivanovo sample.**
(DOCX)

**S2 Table. Pairwise comparison of patients from hyperlipidemia groups by clinical parameters that differ.**
(DOCX)

**S3 Table. Pairwise comparison of maximum stenosis and maximum plaque height.**
(DOCX)

## Author Contributions

**Conceptualization:** Anastasia V. Blokhina, Alexandra I. Ershova, Olga A. Litinskaya.

**Data curation:** Anastasia V. Blokhina, Alexandra I. Ershova, Alexey N. Meshkov.

**Formal analysis:** Anastasia V. Blokhina, Alexandra I. Ershova, Anna V. Kiseleva, Evgeniia A. Sotnikova, Anastasia A. Zharikova, Marija Zaicenoka, Yuri V. Vyatkin, Vasily E. Ramensky, Vladimir A. Kutsenko, Olga A. Litinskaya, Maria S. Pokrovskaya, Svetlana A. Shalnova.

**Funding acquisition:** Alexandra I. Ershova, Svetlana A. Shalnova, Alexey N. Meshkov, Oxana M. Drapkina.

**Investigation:** Anastasia V. Blokhina, Alexandra I. Ershova, Anna V. Kiseleva, Evgeniia A. Sotnikova, Anastasia A. Zharikova, Marija Zaicenoka, Yuri V. Vyatkin, Vasily E. Ramensky, Olga A. Litinskaya, Maria S. Pokrovskaya, Svetlana A. Shalnova.

**Methodology:** Anastasia V. Blokhina, Alexandra I. Ershova, Alexey N. Meshkov.

**Project administration:** Alexandra I. Ershova, Alexey N. Meshkov.

**Resources:** Alexandra I. Ershova, Svetlana A. Shalnova, Alexey N. Meshkov, Oxana M. Drapkina.

**Software:** Anastasia A. Zharikova, Marija Zaicenoka, Yuri V. Vyatkin, Vasily E. Ramensky, Vladimir A. Kutsenko, Olga A. Litinskaya, Maria S. Pokrovskaya, Svetlana A. Shalnova.

**Supervision:** Oxana M. Drapkina.

**Validation:** Anastasia V. Blokhina, Alexandra I. Ershova, Anna V. Kiseleva, Evgeniia A. Sotnikova, Alexey N. Meshkov.

**Visualization:** Anastasia V. Blokhina.

**Writing – original draft:** Anastasia V. Blokhina.

**Writing – review & editing:** Anastasia V. Blokhina, Alexandra I. Ershova, Anna V. Kiseleva, Evgeniia A. Sotnikova, Marija Zaicenoka, Vasily E. Ramensky, Vladimir A. Kutsenko, Alexey N. Meshkov.

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
