## [Decision Letter · Decision Letter 0]

10 Nov 2024

PONE-D-24-30661Clinical and biochemical features of atherogenic hyperlipidemias with different genetic basis: A comprehensive comparative studyPLOS ONE

Dear Dr. Blokhina,

Thank you for submitting your manuscript to PLOS ONE. After careful consideration, we feel that it has merit but does not fully meet PLOS ONE’s publication criteria as it currently stands. Therefore, we invite you to submit a revised version of the manuscript that addresses the points raised during the review process.

We look forward to receiving your revised manuscript.

Kind regards,

Mohammad Reza Mahmoodi, Ph.D.

Academic Editor

PLOS ONE

Journal Requirements:

2. In this instance it seems there may be acceptable restrictions in place that prevent the public sharing of your minimal data. However, in line with our goal of ensuring long-term data availability to all interested researchers, PLOS’ Data Policy states that authors cannot be the sole named individuals responsible for ensuring data access (http://journals.plos.org/plosone/s/data-availability#loc-acceptable-data-sharing-methods).

Reviewers' comments:

Reviewer's Responses to Questions

**Comments to the Author**

1. Is the manuscript technically sound, and do the data support the conclusions?

Reviewer #1: Yes

Reviewer #2: Yes

2. Has the statistical analysis been performed appropriately and rigorously? 

Reviewer #1: Yes

Reviewer #2: Yes

3. Have the authors made all data underlying the findings in their manuscript fully available?

Reviewer #1: No

Reviewer #2: Yes

4. Is the manuscript presented in an intelligible fashion and written in standard English?

Reviewer #1: Yes

Reviewer #2: Yes

5. Review Comments to the Author

Reviewer #1: Blokhina and collaborators study the frequency of CHD using an epidemiological approach that departs from an NSG genetic analysis and a database of 3700 samples from other ongoing studies in the Russian Federation, based on a panel of 6 genes and a number of genetic variants that correspond to lipid disorders of cholesterol metabolism resulting in different clinical/biochemical phenotypes.

Main strengths of the work are the specificity of the distinct phenotypes of hyperlipidemias based on their genetic origin including genetic variants. Among the limitations, also acknowledged by the Authors in the Discussion, are the limited representation of the human population and the low statistical power of the study.

This work illuminates the role of genetics in hyperlipidemias at the origin of phenotypic variability and the difficulties in diagnosing them. The main contribution of this work should be better highlighted from the Abstract. Future work on this important topic should be performed in a more representative sample of the human population.

Main comments

#1. In the Results section, the description of the results and comparisons between the different groups is too long and can be quite shortened since this is repeated in the Discussion. This shortening can be helped by introducing earlier in the text the now Fig. 6. Earlier highlighting of the most salient observations of the study could facilitate a more agile, less tedious, reading of the text.

The Discussion section is okay as well as the comparison of the study's results with other reports.

#2. There is an important mistake in Fig. 5B concerning FD group frequency of CHD; it is not 31% as reported (4/29=13.8 not 31%) half compared to the FH group.

Please, revise because is an important observation and the source of multiple comparisons with the other groups of the experimental design from this study,

Reviewer #2: 1. There are some grammatical, alignments and typographical errors are noted in the manuscript and it should be thoroughly checked and corrected throughout the manuscript.

2. Check the abbreviations throughout the manuscript and introduce the abbreviation when the full word appears the first time in the abstract and the remaining for the text and then use only the abbreviation (For example, TG, etc.,). Make a word abbreviated in the article that is repeated at least three times in the text, not all words to be abbreviated.

3. In materials and methods, the authors should give more detail for control group for better understanding.

4. The authors should provide more information in the figure legends for non-experts. All legends should have enough description for a reader to understand the figure without having to refer back to the main text of the manuscript. For example, the necessary expansion (for abbreviations) should be given which are used in the present investigation.

5. The authors should improve the quality of the images used in the manuscript with high resolution for better understanding. For example, the letters used are blurred and it should be rectified.

6. PLOS authors have the option to publish the peer review history of their article (what does this mean?). If published, this will include your full peer review and any attached files.

Reviewer #1: No

Reviewer #2: **Yes: **Prof. Dr. Vijaya Anand

---

## [Author Response · Author response to Decision Letter 0]

25 Nov 2024

Response to Academic Editor:

The team of authors would like to express their gratitude to the Academic Editor for accepting the manuscript for review, for working with our article, and for all the valuable comments.

A detailed response to each point is provided below.

Journal Requirements

Point 1: Please ensure that your manuscript meets PLOS ONE's style requirements, including those for file naming. The PLOS ONE style templates can be found at

Response 1: 

The revised manuscript meets PLOS ONE's style requirements, including those for file naming.

Unnecessary italics have been removed from the main text (FD line 147, FH line 149, Polygenic HCL line 154, Severe HCL line 157, The control group line 159). Removed italics in symbols "p" and "n" in the whole manuscript. The revised version retains italics and text formatting only where necessary (e.g. genes). 

Corrected font of two level 1 headings of major sections (Limitations line 652, Сonclusion line 671). Updated version has 18pt font for level 1 headings for all major sections.

Corrected the format of supporting information citations: "S1 FigA" line 304, "S1 FigB" line 306.

Revised caption and legend for Figure 4 (Figure 3 of the first version): "Fig 4. Lipid levels among hyperlipidemia groups and compared with the control group. A: LDL-C; B: HDL-C; C: TG; D: Lp(a) levels. Сentral lines represent the median, box limits represent upper and lower quartiles, vertical lines represent 1.5 times the quartile range, and points represent outliers. Horizontal square brackets with p-values above indicate significantly different groups. Only significant differences are indicated among groups. P-values were adjusted by the Holm-Bonferroni method. FD, familial dysbetalipoproteinemia; FH, familial hypercholesterolemia; HCL, hypercholesterolemia; HDL-C, high-density lipoprotein cholesterol; LDL-C, low-density lipoprotein cholesterol; Lp(a), lipoprotein(a); TG, triglycerides".

Revised caption and legend for Figure 6 (Figure 5 of the first version): "Fig 6. Frequency and age at onset of CHD among hyperlipidemia groups compared with the control group. A: age at onset. Central lines represent the median, box limits represent upper and lower quartiles, vertical lines represent 1.5 times the quartile range, and points represent outliers. Only significant differences are indicated among groups. P-values were adjusted by the Holm-Bonferroni method; B: frequency. Horizontal square brackets with p-values above indicate significantly different groups. Only significant differences are indicated among groups. P-values were adjusted by the Holm-Bonferroni method. FD, familial dysbetalipoproteinemia; FH, familial hypercholesterolemia; HCL, hypercholesterolemia; Me, median".

Revised caption and legend for Figure 7 (Figure 6 of the first version): "Fig 7. Severity of carotid and femoral atherosclerosis. A: number of carotid plaques; B: total carotid stenosis; С: carotid plaque score; D: number of femoral plaques; E: total femoral stenosis; F: femoral plaque score. Central lines represent the median, box limits represent upper and lower quartiles, vertical lines represent 1.5 times the quartile range, and points represent outliers. Horizontal square brackets with p-values above indicate significantly different groups. Only significant differences are indicated among groups. For study groups, p-values were obtained from linear regression with adjustments, and from Mann-Whitney test for control vs. study groups. P-values were adjusted by the Holm-Bonferroni method. FD, familial dysbetalipoproteinemia; FH, familial hypercholesterolemia; HCL, hypercholesterolemia; Me, median".

Revised caption and legend for S1 Figure: "S1 Fig. Polygenic risk score in the ESSE-Ivanovo sample (n = 1858). A: LDL-C levels; B: TG levels. The x-axis shows the distribution of the polygenic risk score, and the y-axis shows the number of subjects. Color indicates percentiles".

Corrected the style of the footnote for Table 1: aFor samples: FD (n = 29) and severe HCL (n = 41). Statistical power 80.0%.

Point 2: In this instance it seems there may be acceptable restrictions in place that prevent the public sharing of your minimal data. However, in line with our goal of ensuring long-term data availability to all interested researchers, PLOS’ Data Policy states that authors cannot be the sole named individuals responsible for ensuring data access (http://journals.plos.org/plosone/s/data-availability#loc-acceptable-data-sharing-methods). Data requests to a non-author institutional point of contact, such as a data access or ethics committee, helps guarantee long term stability and availability of data. Providing interested researchers with a durable point of contact ensures data will be accessible even if an author changes email addresses, institutions, or becomes unavailable to answer requests.

Response 2: 

Dear Editor, based on your comments, we have provided additional information in the data availability statement. We have also included non-author contact information: e-mail of the Researcher of the Laboratory of clinomics, Dr. Limonova Alena (limonova-alena@yandex.ru), e-mail and phone number of the Ethics Committee of the National Medical Research Center for Therapy and Preventive Medicine +7 (499)-553-68-10:

The data used in this study, including individual genotype information, cannot be publicly disclosed according to the rules of the Ethics Committee of the National Medical Research Center for Therapy and Preventive Medicine. Deidentified data will be provided upon reasonable request by the corresponding author (blokhina0310@gmail.com) or by the researcher of the laboratory of clinomics, Dr. Limonova Alena (limonova-alena@yandex.ru) or by the Ethics Committee of the National Medical Research Center for Therapy and Preventive Medicine (phone number +7 (499)-553-68-10, SecretaryNEC@gnicpm.ru). Proposals will be reviewed and approved by the investigators, local regulatory authorities, and the Ethics Committee of the National Medical Research Center for Therapy and Preventive Medicine. Once the proposal is approved, data can be transferred through a secure online platform after signing a data access agreement and a confidentiality agreement.

Point 3: Please review your reference list to ensure that it is complete and correct. If you have cited papers that have been retracted, please include the rationale for doing so in the manuscript text, or remove these references and replace them with relevant current references. Any changes to the reference list should be mentioned in the rebuttal letter that accompanies your revised manuscript. If you need to cite a retracted article, indicate the article’s retracted status in the References list and also include a citation and full reference for the retraction notice.

Response 3: 

All references were checked for style and corrected. In the revised manuscript, we also added the one additional reference to our study related to autosomal dominant familial dysbetalipoproteinemia:

"19. Blokhina AV, Ershova AI, Kiseleva AV, Sotnikova EA, Zharikova AA, Zaicenoka M, et al. Spectrum and Prevalence of Rare APOE Variants and Their Association with Familial Dysbetalipoproteinemia. Int J Mol Sci. 2024; 25:12651. doi: 10.3390/ijms252312651".

Therefore, references have been renumbered.

Response to Reviewers comments

We would like to express our sincere gratitude to the reviewers for their constructive comments, professionalism, and interest in our study. We greatly appreciate the time the reviewers spent reviewing our article.

Response to Reviewer 1 comments:

Point 1: In the Results section, the description of the results and comparisons between the different groups is too long and can be quite shortened since this is repeated in the Discussion. This shortening can be helped by introducing earlier in the text the now Fig. 6. Earlier highlighting of the most salient observations of the study could facilitate a more agile, less tedious, reading of the text.

The Discussion section is okay as well as the comparison of the study's results with other reports.

Response 1: 

We sincerely appreciate the reviewer's comments, which have prompted us to make several additions to the Results and Discussion section.

We believe that the reviewer was referring to Figure 7 (Overview of study findings), which summarizes the results of the study. We agree with the comment. Figure 7 (Overview of study findings) has been moved to the beginning of the Results section (before the comparison of key data, line 403, Fig 3. of the current version). An earlier presentation of the key findings will make it easier to read the entire Results section.

Several adjustments have also been made to the Results and Discussion section. In particular, we have removed unnecessary repetition:

In the Results section, we have removed repetitive text that has already been presented in the figures (lines 429-430, 441-442, 480, 487-488).

Discussion section:

From the subsection Lipid levels:

"We performed a simultaneous comparison of lipid levels in patients with genetically-based hyperlipidemias (FH, FD, and polygenic HCL), severe HCL, and subjects without lipid disorders". 

From the subsection Frequency and age at onset of coronary heart disease:

"We conducted a comparative analysis of the frequency and age at onset of CHD, which includes myocardial infarction and coronary artery revascularization, in patients with genetically-based hyperlipidemias, severe HCL, and subjects without lipid disorders".

From the subsection Severity of carotid and femoral atherosclerosis subtitle:

"…in patients with genetically-based hyperlipidemias, severe HCL, and subjects without lipid disorder".

The abstract has also been improved.

Point 2: There is an important mistake in Fig. 5B concerning FD group frequency of CHD; it is not 31% as reported (4/29=13.8 not 31%) half compared to the FH group. Please, revise because is an important observation and the source of multiple comparisons with the other groups of the experimental design from this study.

Response 2: 

We sincerely thank Reviewer 1 for this comment. Indeed, there was a typo in Fig 5B: 9 patients with FD (not 4 patients) had the coronary heart disease. We sincerely apologize for this typo. This typo has been corrected. The frequency of coronary heart disease (31.0%) is correct, as are all related statistical data analyses.

Response to Reviewer 2 comments:

Point 1: There are some grammatical, alignments and typographical errors are noted in the manuscript and it should be thoroughly checked and corrected throughout the manuscript.

Response 1: 

Grammatical, alignment, and typographical errors have been thoroughly checked and corrected throughout the manuscript. All changes are highlighted in yellow. 

Point 2: Check the abbreviations throughout the manuscript and introduce the abbreviation when the full word appears the first time in the abstract and the remaining for the text and then use only the abbreviation (For example, TG, etc.,). Make a word abbreviated in the article that is repeated at least three times in the text, not all words to be abbreviated.

Response 2: 

All abbreviations in the manuscript were checked. We have included abbreviations in the article if they are repeated at least three times in the text. Unnecessary abbreviations have been removed. All changes are highlighted in yellow.

Point 3: In materials and methods, the authors should give more detail for control group for better understanding.

Response 3: 

Improved description of inclusion criteria for subjects in the control group in the Materials and methods (lines 159-163):

"The control group consisted of subjects without lipid disorders from the population-based sample (ESSE-Ivanovo). These subjects had no causal variants in 25 genes associated with lipid disorders included in the custom panel analyzed (see NGS section in Materials and methods) and had a low polygenic risk score for LDL-C (< 50th percentile). Their LDL-C levels were < 3.0 mmol/L and TG levels were < 1.5 mmol/L (n = 144)".

Point 4: The authors should provide more information in the figure legends for non-experts. All legends should have enough description for a reader to understand the figure without having to refer back to the main text of the manuscript. For example, the necessary expansion (for abbreviations) should be given which are used in the present investigation.

Response 4: 

 All figure descriptions have been improved in accordance with reviewer comments. In particular, we have added more information to figure legends and expanded abbreviations. All changes are highlighted in yellow:

Fig 1 (lines 169-172), Fig 2 (lines 265-268), Fig 3 (lines 403-412), Fig 4 (lines 418-427), Fig 5 (lines 452-457), Fig 6 (lines 465-473), Fig 7 (lines 497-506).

 The legend for Table 2 has also been improved (lines 369-379).

Point 5: The authors should improve the quality of the images used in the manuscript with high resolution for better understanding. For example, the letters used are blurred and it should be rectified.

Response 5: 

In accordance with PLOS requirements, the quality of all figures in the main text and supplementary figures has been improved. Each updated figure has a high resolution. The PACE (Preflight Analysis and Conversion Engine) digital diagnostic tool was also used.

---

## [Decision Letter · Decision Letter 1]

29 Nov 2024

Clinical and biochemical features of atherogenic hyperlipidemias with different genetic basis: A comprehensive comparative study

PONE-D-24-30661R1

Dear Dr. Blokhina,

We’re pleased to inform you that your manuscript has been judged scientifically suitable for publication and will be formally accepted for publication once it meets all outstanding technical requirements.

Kind regards,

Mohammad Reza Mahmoodi, Ph.D.

Academic Editor

PLOS ONE

Additional Editor Comments (optional):

Reviewers' comments:

Reviewer's Responses to Questions

**Comments to the Author**

1. If the authors have adequately addressed your comments raised in a previous round of review and you feel that this manuscript is now acceptable for publication, you may indicate that here to bypass the “Comments to the Author” section, enter your conflict of interest statement in the “Confidential to Editor” section, and submit your "Accept" recommendation.

Reviewer #1: All comments have been addressed

Reviewer #2: All comments have been addressed

2. Is the manuscript technically sound, and do the data support the conclusions?

Reviewer #1: Yes

Reviewer #2: Yes

3. Has the statistical analysis been performed appropriately and rigorously? 

Reviewer #1: Yes

Reviewer #2: Yes

4. Have the authors made all data underlying the findings in their manuscript fully available?

Reviewer #1: Yes

Reviewer #2: Yes

5. Is the manuscript presented in an intelligible fashion and written in standard English?

Reviewer #1: Yes

Reviewer #2: Yes

6. Review Comments to the Author

Reviewer #1: (No Response)

Reviewer #2: 1. The authors may include different lipid ratio (cholesterol ratio) for better understaning and outcome of the manuscript.

7. PLOS authors have the option to publish the peer review history of their article (what does this mean?). If published, this will include your full peer review and any attached files.

Reviewer #1: No

Reviewer #2: **Yes: **Prof. A. Vijaya Anand

---

## [Editor Report · Acceptance letter]

10 Dec 2024

PONE-D-24-30661R1 

PLOS ONE

Dear Dr. Blokhina, 

I'm pleased to inform you that your manuscript has been deemed suitable for publication in PLOS ONE. Congratulations! Your manuscript is now being handed over to our production team.

Kind regards, 

on behalf of

Dr. Mohammad Reza Mahmoodi 

Academic Editor

PLOS ONE